# Genetic Algorithm-Back Propagation Neural Network Model- and Response Surface Methodology-Based Optimization of Polysaccharide Extraction from *Cinnamomum cassia* Presl, Isolation, Purification and Bioactivities

**DOI:** 10.3390/foods14040686

**Published:** 2025-02-17

**Authors:** Qicong Chen, Wenqing Zhang, Yali Wang, Weifeng Cai, Qian Ni, Cuiping Jiang, Jiyu Li, Chunyan Shen

**Affiliations:** 1School of Traditional Chinese Medicine, Southern Medical University, Guangzhou 510515, China; cqc18529114382@163.com (Q.C.); lily990826@163.com (Y.W.); cwf752651751@163.com (W.C.); 15116565861@163.com (Q.N.); cuipingjiangcpu@163.com (C.J.); 2College of Engineering, South China Agricultural University, Guangzhou 510642, China; 15378770939@163.com (W.Z.); lijiyu@scau.edu.cn (J.L.)

**Keywords:** anti-inflammation, artificial neural network, *Cinnamomum cassia*, optimization, polysaccharide

## Abstract

Ultrasonic-assisted enzymatic extraction (UAEE) was utilized to obtain the polysaccharides from the bark of *Cinnamomum cassia* Presl. (*C. cassia*). Taking the yield of the crude polysaccharides from *C. cassia* (CCCP) as the assessment indicator, response surface methodology (RSM) and a genetic algorithm-back propagation (GA-BP) artificial neural network model were employed to forecast and contrast the optimal parameters for UAEE. The outcomes demonstrated that the GA-BP model, which was superior in prediction accuracy and optimization capabilities to the RSM and BP models, identified the following conditions as optimal for the UAEE of CCCP: cellulase was employed, the temperature for enzymatic hydrolysis was 50.0 °C, the pH value was 5.248, the addition of enzyme was 3%, and the ultrasonic time was 70.153 min. Under these parameters, the yield of CCCP was significantly increased to 28.35%. Then, UAEE-extracted CCCP under optimal conditions was further separated and purified using a DEAE-52 column and SephadexG-100 column, yielding five purified polysaccharides from *C. cassia* (CCPs). All of these five fractions were acidic polysaccharides with safety at 3 mg/mL. The CCPs did not significantly affect the viability of HaCaT cells affected by UVB exposure. The CCPs demonstrated differential inhibition of nitric oxide production in RAW264.7 cells stimulated by lipopolysaccharide.

## 1. Introduction

Polysaccharides, owing to their various biological activities, including antioxidant [1], anti-cancer [2], and immunomodulatory [3] effects, have been extensively applied in the field of food and biomedical applications [4,5]. As macromolecular components, polysaccharides are typically extracted using the water extraction method due to their property of being soluble in water but insoluble in ethanol [6]. This approach is straightforward in principle and easy to operate. However, it might damage the structure of plant polysaccharides because of the requirements of high temperature and long extraction time. Hence, it is essential to develop sustainable and effective techniques for the extraction of polysaccharides.

Ultrasonic extraction has gradually gained widespread recognition for polysaccharide extraction [7]. Especially, ultrasonic-assisted enzymatic extraction (UAEE) has shown excellent performance in extracting polysaccharides from plants [6]. Jong Jin Park et al. [8] compared the extraction methods of hot buffer, ultrasound, enzymatic methods, and a combination of ultrasound with enzymes and found that UAEE exhibited more efficient extraction efficiency and maintained the structure of polysaccharides to the greatest extent possible to guarantee their activity. When the ultrasonic process is used, the cavitation phenomenon will break down the cell wall, causing the leakage of cellular contents. This is beneficial to the entry of enzymes and the target compound or its substrate and greatly improves the extraction yield [9]. However, there are also some drawbacks to the UAEE method, such as the possibility of damaging the structure of polysaccharides or making it difficult to separate polysaccharides during the extraction process, which may affect the yield [6]. Therefore, optimizing the UAEE method to improve the polysaccharide yield is particularly important.

The optimization of polysaccharide extraction processes mainly adopts the response surface method (RSM) [10]. However, there are complex non-linear relationships among the extraction conditions during the extraction process and traditional methods have certain limitations in dealing with non-linear problems [11]. A back propagation (BP) artificial neural network (ANN) is composed of processing neurons that are highly interconnected by linear or non-linear transfer functions, which can achieve multiple linear regression of weights and biases [12]. Combined with a genetic algorithm (GA) that searches for the best extraction plan through natural selection and genetic mechanisms, the GA-BP model, as a complementary model superior to RSM, has attracted increasing attention from scholars performing research on optimizing extraction conditions [13].

Cinnamon is the dried bark of *Cinnamomum cassia* Presl (*C. cassia*), which is mainly produced in China and Southeast Asian regions. The raw material of *C. cassia* possesses complex chemical compositions. The predominant components encompass the most abundant chemical components, namely terpenoids, such as monoterpenes (e.g., d-limonene, α-terpineol), sesquiterpenes (e.g., caryophyllene, α-cubebene), and diterpenes (e.g., cinnamyl alcohol, dehydrocinnamyl alcohol), highly active and abundant phenylpropanoids, such as cinnamaldehyde and cinnamic acid, along with certain glycosides. The secondary components consist of other compounds, like lignins, lactones, and proteins [14]. Polysaccharides, as one of the major constituents of *C. cassia*, have attracted great attention because of their beneficial effects. For example, Al-Ajalein AAS et al. discovered that cinnamon polysaccharides possess excellent capabilities in protecting the skin barrier and preventing pigmentation [10]. However, the majority of reports regarding polysaccharide extraction focused on conventional methods, including pressurized hot water extraction and distillation extraction [15,16]. The traditional extraction method of water extraction followed by alcohol precipitation imposes significant limitations on the acquisition and extensive application of polysaccharides from *C. cassia*. Recently, modern extraction techniques such as ultrasonic-assisted extraction, subcritical water extraction, and microwave-assisted extraction have been explored to improve the extraction efficiency and yield of polysaccharides. A low-molecular-weight pectin polysaccharide was isolated from cinnamon bark using microwave-assisted extraction, with a yield of 13% [10]. Therefore, it is necessary to optimize the extraction process of polysaccharides from *C. cassia* using a more environmentally friendly and efficient extraction process. In the current study, UAEE was utilized to extract polysaccharides from *C. cassia* and the extraction conditions were optimized. The anti-inflammatory activity and photodamage protection activity of the polysaccharides from *C. cassia* extracted and isolated under these conditions were evaluated, indicating that the UAEE method was efficient in extracting and maintaining the biological activity of the polysaccharides. This study provides a scientific basis for the potential applications of polysaccharides from *C. cassia* and UAEE in various fields.

## 2. Materials and Methods

### 2.1. Extraction of Crude Polysaccharides from C. cassia (CCCP)

Cinnamon powder was enzymatically hydrolyzed for 30 min at 50 °C utilizing a citric acid-sodium citrate buffer (pH 5.0 ± 0.05) containing 0.03% cellulase. The mixture was sonicated for 90 min in an ultrasonic device with an output power of 100 W (KQ-100, Kunshan Ultrasonic Instrument Co., Ltd., Kunshan, China) and subsequently heated at 90 °C for 5 min to deactivate enzymes. The filtrate was concentrated and decolorized with macroporous weakly alkaline anion resin (D354FD, Zhejiang Zhengguang Industrial Co., Ltd., Hangzhou, China) and precipitated with anhydrous ethanol overnight at 4 °C. The following day, the precipitate was collected by centrifugation at 8000 rpm for a duration of 10 min. Subsequently, the polysaccharides were deproteinized using the sevage method, precipitated by anhydrous ethanol, and lyophilized to obtain the crude polysaccharides from *C. cassia*, which were named as CCCP.

### 2.2. Single-Factor Experiment

With the yield of CCCP as the evaluation index, univariate experiments were conducted on UAEE for CCCP, focusing on single-variable experiments (type of enzyme, addition of enzyme, ultrasonic time, pH value, and enzymatic temperature). Each experiment concentrated on changing a single factor while maintaining the other variables at a constant level: the enzyme type was cellulase, the enzyme addition was 3.0%, the ultrasonic time was 30 min, the pH value was 5, and the ultrasonic temperature was 50 °C. The experimental conditions for the single-factor experiment were as follows: enzyme type included cellulase: pectinase ratios of 1:1, 1:2, 2:1, cellulase alone, and pectinase alone; addition of enzyme was 0.5%, 1.5%, 3.0%, 4.5%, and 6.0%; ultrasonic time was 20, 40, 60, 80, and 100 min; pH values were 4, 5, 6, and 7; and ultrasonic temperature was 30, 40, 50, 60, and 70 °C. The yield of CCCP was calculated using the following equation:(1)$$Yield\left(\%\right)=\frac{m1}{m2}\times 100\%$$
where m1 denotes the mass of CCCP after drying, and m2 indicates the mass of the raw material.

### 2.3. Response Surface Methodology (RSM)

According to the findings from the single-factor experiments, the chosen factors included pH value (A), ultrasonic time (B), and addition of enzyme (C), adhering to the principles of Box-Behnken design. Table 1 presents the comprehensive experimental protocol of the response surface methodology.

### 2.4. ANN Modeling

#### 2.4.1. Back Propagation (BP) Neural Network

The non-linear relationship between the independent variables and the responses was modeled using MATLAB R2024a (MathWorks Inc., Natick, MA, USA) [13]. The network architecture consisted of one input layer (encompassing the pH value, addition of enzyme, and ultrasonic time), one hidden layer (comprising ten hidden layer neurons), and one output layer (representing the yield of CCCP extraction). The specific topological structure is depicted in Figure 1A. In the experiment, the quantity of neurons in the hidden layer was determined in accordance with the Kolmogorov theorem, as expressed in Formula (2):(2)$$n2=\sqrt{n1+m+1}+a$$
wherein *n* denotes the number of neurons in the hidden layer, m represents the number of nodes in the input layer, and a is a constant ranging between 1 and 10.

Through testing, it was discovered that when the number of neurons in the hidden layer reached 10, the training accuracy of the network satisfied the requirements. The complete dataset from 17 extraction experiments was divided into three subsets. Specifically, 70% of the RSM design was used for the training dataset, 15% of the RSM design was allocated to validation, and another 15% was used for the testing dataset. A multilayer feed-forward neural network, which was trained by an error BP algorithm, was employed to model the target responses [12].

#### 2.4.2. Genetic Algorithm Optimization in BP Neural Networks

A genetic algorithm (GA) was utilized to conduct further optimization of the BP neural network. The non-linear mapping relationship established by the BP neural network was adopted as the fitness function for the GA [12,13]. Precisely, the data generated by the BP neural network were utilized as the initial population. GAs emulate natural phenomena, including species reproduction, hybridization, mutation, competition, and selection, to iteratively adjust the population. The objective of this process was to optimize the output variable to reach its maximum value. With respect to the three factors, namely pH value, ultrasonic time, and enzyme addition amount, within the extraction conditions, ANN simulations were carried out within designated ranges. Global optimization was attained by formulating an individual fitness function based on the model fitting values [17]. The specific flowchart of the GA-BP neural network is shown in Figure 1B.

#### 2.4.3. Comparison of RSM, BP, and GA-BP Neural Network Model Validation

To validate the precision of the obtained statistical outcomes, an additional series of experimental condition combinations was implemented. Subsequently, both the predicted data and the experimental data were carefully analyzed and compared to assess the validity of the polynomial model.

The coefficient of determination (R^2^), the root mean squared error (RMSE), and the average absolute deviation (AAD) were calculated to evaluate and compare the predictive proficiency and estimation potential of the RSM, BP, and GA-BP neural network models.

### 2.5. Isolation and Purification of CCCP

CCCP was loaded onto a DEAE-52 column (3.5 × 50 cm, S14024, Shanghai Yuanye Bio-Technology Co., Ltd., Shanghai, China) and successively eluted using distilled water along with NaCl solutions at varying concentrations of 0.05, 0.1, 0.3, and 0.5 mol/L. Five fractions were collected based on the elution profiles from the phenol-sulfuric acid assay, dialyzed against distilled water for 48 h, and lyophilized [18]. These five fractions underwent purification using a Sephadex G-100 (S14034, Yuanye, Shanghai, China) column measuring 1.5 × 100 cm, with distilled water passed through with a flow rate set at 0.33 mL/min, during which the elution profiles were monitored and analyzed.

### 2.6. Determination of Chemical Compositions

The total carbohydrate content was quantified using the phenol-sulfuric acid method [18]. Specifically, to a 2 mL sample solution, 0.05 mL of 80% phenol solution was added. After thorough mixing, 5.0 mL of concentrated sulfuric acid was rapidly introduced. The reaction was maintained for 10 min and then cooled to room temperature. The absorbance was measured at 490 nm, and a standard curve was plotted using glucose (G116305, Aladdin, Shanghai, China) as the standard to calculate the total carbohydrate content.

The protein concentration was assessed using the Bradford assay [19]. Specifically, 100 mg of Coomassie Brilliant Blue G-250 (C.I.42655, Sigma-Aldrich, Saint Louis, MO, USA) was dissolved in 50 mL of 95% ethanol, mixed with 100 mL of 85% phosphoric acid, diluted to 1000 mL, and then filtered. Next, 5 mL of Coomassie Brilliant Blue G-250 solution was added respectively to 1 mL of BSA solutions (0–1.0 mg/mL) and the sample solutions. After incubation at room temperature for 15 min, and the absorbance was measured at 595 nm. The protein content was calculated based on the BSA standard curve.

Moreover, the starch content was evaluated using the iodine titration method [20]. The galactosan content was assessed using the *meta*-hydroxybiphenyl (S30798, Yuanye, Shanghai, China) method at 525 nm [21].

### 2.7. Gel Permeation Chromatography (GPC) Analysis

The molecular weight, uniformity, and polydispersity index of CCP1, CCP2, and CCP3 were characterized using GPC. The GPC system comprised a Shimadzu LC-20A instrument (Shimadzu, Kyoto, Japan) equipped with an Ultra hydrogel Linear column (7.8 mm × 300 mm, Waters, Milford, MA, USA) and a refractive index detector (RID) (model 2414, Waters, Milford, MA, USA). Sample preparation involved dissolving the samples at a concentration of 1 mg/mL in distilled water, followed by filtration through a 0.22 μm filter. Subsequently, 50 μL of the filtered solution was injected into the GPC system at 35 °C. Elution was performed using 0.1 M sodium nitrate as the mobile phase at a flow rate of 0.5 mL/min. Data acquisition was monitored using a Waters 2414 differential refractometer, and data processing was conducted using Breeze2 GPC software (Breeze 2, Waters, Milford, MA, USA).

### 2.8. Structural Characterization

CCP1, CCP2, and CCP3 along with KBr were dried under an infrared lamp for approximately 2 h. The samples were finely ground and homogenized with 200 mg of pure KBr and then pressed into thin wafers on a tablet press. Infrared scanning was conducted within the wavenumber range of 4000–400 cm^−1^ using a Fourier transform infrared spectrophotometer (Thermo Nicolet iS50 FT-IR, Waltham, MA, USA).

### 2.9. Erythrocyte Hemolysis Test

Blood was collected from SD rats via abdominal aorta puncture into tubes containing anticoagulant. Following the addition of an equal volume of 0.9% sodium chloride solution and centrifugation at 2500 rpm for 5 min, the supernatant was removed. The obtained red blood cells were resuspended in physiological saline to produce a 2% (*v*/*v*) suspension. The tested samples (3 mg of each) were dissolved in 2.5 mL of 0.9% sodium chloride solution. Both the cell suspensions and sample solutions were centrifuged at 2000 rpm for 10 min. The absorbance of the supernatants was recorded at 545 nm with a UV spectrophotometer, using distilled water as a blank.

### 2.10. Cell Culture

HaCaT and RAW264.7 cells were obtained from the Chinese Academy of Sciences Cell Bank (Shanghai, China). Dulbecco’s Modified Eagle Medium (DMEM) supplemented with 100 IU/mL of penicillin, 100 μg/mL of streptomycin, and 10% (*v*/*v*) fetal bovine serum was used to culture the cells. The cells were kept at 37 °C in a humidified incubator under a 5% CO_2_ atmosphere.

### 2.11. Cell Viability Assay

MTT (3-(4,5-dimethylthiazol-2-yl)-2,5-diphenyltetrazolium bromide) is a yellow tetrazolium salt used to assess cell viability and proliferation. In metabolically active cells, mitochondrial succinate dehydrogenase reduces MTT to insoluble purple formazan crystals.

HaCaT and RAW264.7 cells were plated in 96-well plates at a density of 5 × 10^3^ cells per well. After incubation for 12 h, the cells were treated with CCP1 (6.25–100 μM), CCCP, CCP2, CCP3, CCP4, and CCP5 (31.25–500 μM) dissolved in DMSO (≤0.1%) for 24 h. After adding MTT, the cells were incubated for another 4 h, followed by the addition of 150 μL of DMSO. Cell viability was evaluated by determining the absorbance at 570 nm.

For the UVB irradiation studies, HaCaT cells were subjected to different UVB doses for 24 h (0, 15, 30, 60, and 120 mJ/cm^2^). Then, the viability of the HaCaT cells was subsequently evaluated using the MTT test to determine the effects of UVB irradiation. To test the effects of polysaccharides from *C. cassia* (CCPs, including CCCP, CCP1, CCP2, CCP3, CCP4, and CCP5) on UVB-induced HaCaT cells, cells were seeded and irradiated at 30 mJ/cm^2^ and then exposed to the appropriate concentrations of the CCPs for 24 h, after which the MTT assay was conducted to assess cell viability.

### 2.12. Nitric Oxide (NO) Production Assay

RAW264.7 cells were seeded in 24-well plates at 2 × 10^5^ cells/well. After attachment, the Con and LPS groups received DMEM (containing 0.1% DMSO), while the other groups were given various concentrations of the CCPs or 50 μg/mL of dexamethasone (DXM). After 24 h, the cells in the control group were given fresh medium, while other groups were exposed to 1 μg/mL of LPS for another 24 h. After treatment, the culture media were collected, and the NO content was measured at 540 nm using the Griess reagent kit assay (Beyotime Biotechnology S0021, Beyotime Biotechnology, Shanghai, China).

### 2.13. Statistical Analysis

Statistical analyses were carried out with SPSS version 25.0 (IBM Corp., Armonk, NY, USA). Differences among groups were evaluated through one-way ANOVA, followed by Tukey’s post hoc test for pairwise comparisons. Significance levels of *p* < 0.05, *p* < 0.01, and *p* < 0.001 were established to indicate statistically significant differences. The comparison was made against the CON group for *p*-values.

## 3. Results and Discussion

### 3.1. Impact of Various Factors on UAEE Efficiency

As illustrated in Figure 2A, when the type of enzyme was cellulase, the yield of CCCP was the highest. This might be owing to the fact that cellulase could decompose the cell wall of cinnamon more effectively than pectinase, enhancing the leaching rate of the active components [22]. Hence, the type of enzyme employed in the subsequent experiments was set as cellulase.

Figure 2B shows that the yield of CCCP displayed an increasing trend as the enzyme dosage was gradually raised. When the enzyme concentration was raised to 4.5%, the yield of CCCP reached its peak at 24.21%, which was significantly higher than that of 17.72% at an enzyme dosage of 1.5%. One possible reason was that before the enzyme dosage reached 4.5%, the contact opportunities between the substrate and the enzyme increased with the rise in the enzyme dosage, thereby facilitating the release of polysaccharides [23]. Nevertheless, once the addition of enzyme was excessively high, various components would mutually inhibit each other, resulting in a reduction in the polysaccharide yield [6]. Hence, the enzyme dosage employed in the subsequent experiments was chosen as 4.5%.

The yield of CCCP ascended within a certain range with elongation of the ultrasonic treatment duration (Figure 2C). The cavitation effect generated by ultrasound increased the contact efficiency of cellulase with the cell walls of cinnamon, thus enhancing the extraction efficiency of CCCP [9]. After 60 min of ultrasonic treatment, the yield of CCCP attained its maximum value and then gradually declined. This was probably attributed to the elongation of the ultrasonic time, which caused transient high temperature and release of free radicals, thereby leading to the degradation of polysaccharides [24]. Consequently, the ultrasonic time was set at 60 min in the subsequent experiments.

Figure 2D demonstrates that the yield of CCCP increased within a certain range as the pH shifted from acidic to neutral. When the pH reached 5.0, the yield of CCCP reached a maximum of 20.57%, which was approximately 5-fold higher than that at pH = 4, and subsequently decreased gradually. Under different pH conditions, the dissociation status of the substrate (such as polysaccharide molecules) underwent alterations, influencing the interaction between the enzyme and the substrate. The results indicated that the optimal pH for cellulase was 5.0. In support, Deepa Deswal et al. also found that cellulase had optimal activity within this pH range [25]. Thus, the pH employed in the subsequent experiments was set at 5.0.

As depicted in Figure 2E, the outcomes suggested that with the elevation of the enzymatic hydrolysis temperature, the yield of CCCP gradually rose. Nevertheless, when the temperature reached 50 °C, the upward tendency started to decelerate. As the temperature ascended, the enzyme activity could increase, leading to a higher release rate of polysaccharides [23]. However, once the temperature exceeded the enzyme’s optimal range, its activity might decrease, leading to sluggish growth in the polysaccharide yield [26]. When the temperature was increased to 60 °C, the yield of CCCP was similar to that at 50 °C, which were 23.15% and 23.16%, respectively. Simultaneously, there were probably some temperature-dependent structural changes in the polysaccharides that could cause a gradual increase in the polysaccharide yield during the plateau phase. Therefore, in the subsequent experiments, considering the energy loss as well as the structural stability of the polysaccharides, the temperature for enzymatic hydrolysis was established at 50 °C.

### 3.2. Optimization of the Extraction Conditions of UAEE

#### 3.2.1. RSM Model Fitting and Statistical Evaluation

According to the experimental outcomes of the single-factor analysis, three factors (pH value, addition of enzyme, and ultrasonic time) were chosen for further optimization. The RSM design matrix was generated using Design Expert 12 software, which is listed in Table 2 together with the corresponding predicted values. By conducting multiple regression analysis, the correlation between the response values and the variables was represented by Equation (3), as follows:Y = 27.522 + 4.575A + 1.95B + 1.665C + 0.730AB + 1.660AC − 0.170BC − 7.6810A^2^ − 3.511B^2^ − 3.3410C^2^(3)

(A: pH; B: addition of enzyme; C: ultrasonic time; Y: yield of CCCP)

The outcomes of the statistical analysis are presented in Table 3. The F value, which was adopted to evaluate the overall significance of the model, yielded a value of 3.84, and the R^2^ value was found to be 0.8358, suggesting that the model was meaningful. In this case, the pH value was the significant model term. However, its interaction with other factors was not significant. The F value of lack-of-fit was recorded at 3.03 and the *p*-value was 0.1506 > 0.05, indicating that it was insignificant in comparison to pure error. A non-significant lack of fit indicated a favorable situation, as it implies that the model was capable of providing a satisfactory fit. Therefore, this model demonstrated a strong goodness of fit. To sum up, the model was successfully established and was of statistical significance [9,23,24,25].

Based on the data in Table 3, the polysaccharide extraction rate was affected by various factors in the following order: pH (F = 9.41) > addition of enzyme (F = 1.71) > ultrasonic time (F = 1.25). Evidently, pH exerted the most significant influence on the change in the extraction rate of CCCP.

By means of the regression equation, in conjunction with the three-dimensional response surface graphs and their associated contour diagrams that were generated by Design Expert software, the interactive relationships of each factor on the extraction rate of CCCP and their influence on the optimal extraction plan could be visually presented [23,25]. The contour plots reflected the degree of interaction among the factors, where an elliptical trend indicated a significant interaction (Figure 3A,B), while a circular trend indicated an insignificant interaction (Figure 3C). In the response surface plots, the higher the steepness of the surface, the greater the influence on the extraction rate and the more significant the influence of factor interaction (Figure 3D,E). Conversely, a flatter surface indicated that the interaction among factors had less impact on the extraction rate (Figure 3F). The curve in Figure 3G is the projection of a two-dimensional surface in three-dimensional space, and its curvature can reflect the strength of the non-linear effect. The curve indicated that pH had the greatest influence on the extraction rate of CCCP, while the interaction between enzymatic hydrolysis temperature and ultrasonic time had a lesser degree of influence on the extraction rate. The stability of the model was further verified by scatter plots of the predicted values versus actual values (Figure 3H), Cook’s distance (Figure 3I), and normal probability (Figure 3J).

Based on the response surface experiment and the data analysis in Design Expert software, the optimal process parameters for the UAEE of CCCP were ultimately determined as follows: cellulase was employed, the temperature for enzymatic hydrolysis was 50.0 °C, the pH during the process was 5.997, the addition of enzyme was 3.825%, the ultrasonic time was 79.895 min, and the theoretical yield was 27.95%.

#### 3.2.2. ANN Model Fitting and Statistical Evaluation

The data obtained from the RSM experiment and single-factor experiments were randomly selected with a total of 33 groups and divided into a test set (accounting for 70%, shown in Figure 4A), a prediction set (accounting for 15%, shown in Figure 4B), and a calibration test set (accounting for 15%, shown in Figure 4C) to construct a three-layer BP-NN model. In the scatter plot of the target values of the BP model set (Figure 4D), the fitness values of the BP neural network were obtained by comparing the experimental values, training values, and validation values. It can be seen from this figure that the regression coefficients of the developed training, testing, and validation models were 0.94179, 0.93471, and 0.99364, respectively, and the overall model regression coefficient was 0.9521. According to the training results of the BP neural network, the R value was approximately close to 1. Such a high regression value indicated that the predicted values had a high similarity with the actual values of the dataset, which fully demonstrated that the constructed neural network model had a reliable prediction ability for the experimental results [17]. As the number of model training steps continued to increase, the MSE gradually decreased and approached the optimal value. At the 3th epoch, the neural network reached the optimal validation performance, and its MSE was 0.003866 (Figure 4E). The total error range of the neural network was from −0.1754 (the leftmost bin) to 0.2344 (the rightmost bin), and the error of the model set fluctuated near zero error (Figure 4F). All of these results indicated that the training convergence speed of the neural network was relatively fast and extremely stable, and the established model could be used in subsequent experimental analyses [12].

Subsequently, the final optimal BP neural network mentioned above was input to generate the best fitness function and the genetic algorithm (GA) was used to find the optimal solution of extraction condition. During the cyclic iterative operation process of crossover-selection-mutation-crossover-selection, with the continuous iterative process, the rapid decrease in the MSE value showed that the model could quickly learn and improve its performance in the initial stage of training [13]. When the number of iterations reached 600, the GA stopped running and output the individual with the highest CCCP yield. The finally obtained optimal process parameters were as follows: cellulase was employed, the temperature for enzymatic hydrolysis was 50.0 °C, the pH during the process was 5.248, the addition of enzyme was 3%, the ultrasonic time was 70.153 min, and the theoretical yield was 28.42%.

#### 3.2.3. Validation and Contrast Among RSM, BP, and GA-BP Neural Network Models

The RSM and ANN (encompassing BP and GA-BP neural networks) models were comparatively analyzed based on the R^2^, RMSE, and AAD, and the results are presented in Table 4. The determination coefficient values of the RSM, BP, and GA-BP models all exceeded 0.83, signifying that the overall fitting performance of these models was relatively satisfactory. Notably, the RMSE and AAD of the GA-BP neural network model were lower than those of the RSM and BP neural network models (Figure 5A). Consequently, the predicted values obtained from the model fitting exhibited a reduced degree of dispersion, thereby leading to more accurate prediction outcomes (Figure 5B).

In light of the fact that the established GA-BP neural network model exhibited greater fitting precision, accuracy, and predictive power compared to the RSM model, the reliability of the former was further scrutinized (Figure 5C). The optimal combinations and predicted values optimized by the RSM, BP, and GA-BP neural network models were verified and contrasted. The results are shown in Table 5. The relative error values of all three models were less than 5%. However, the GA-BP neural network model possessed the lowest MSE, which implied that the GA-BP neural network model boasted greater precision and applicability. The enhanced accuracy of the ANN approach might be attributed to its inherent universal capacity to approximate nonlinear systems. By contrast, the RSM model was founded solely on second-order polynomial regression [12,13,17]. To sum up, in comparison with the RSM and BP neural network models, the GA-BP model demonstrated a superior prediction effect regarding the extraction of CCCP via ultrasonic-assisted enzymatic hydrolysis. Nevertheless, regardless of whether the RSM, BP, or GABP neural network models was employed, the extraction efficiency of CCCP by the optimized UAEE was superior to the previously reported microwave-assisted extraction with an efficiency of 13% [10]. This laid a foundation for the application of UAEE in the extraction of CCCP.

### 3.3. Analysis of the Results of CCCP Isolation and Purification

CCCP was isolated and purified via DEAE-52 column chromatography, with gradient elution performed sequentially using distilled water and NaCl solutions of various concentrations. During this process, the concentration of purified polysaccharides was monitored in real time using the phenol-sulfuric acid method to obtain the elution curve of the DEAE-52 column (Figure 6A). The results showed that a total of five cinnamon purified polysaccharide fractions were obtained, corresponding to the elution fractions with distilled water, 0.05 M, 0.1 M, 0.3 M, and 0.5 M NaCl solutions, which were named CCP1, CCP2, CCP3, CCP4, and CCP5, respectively.

The purified polysaccharides underwent purification using Sephadex G-100 column chromatography, which were eluted with distilled water. During this process, the content of polysaccharides was measured in real time to obtain the elution curve from gel column chromatography. As shown, three single and symmetrically sharp peaks were obtained when CCP1 (Figure 6B), CCP2 (Figure 6C), and CCP3 (Figure 6D) were purified using Sephadex G-100 column chromatography, while CCP4 (Figure 6E) and CCP5 (Figure 6F) were not, implying that CCP1, CCP2, and CCP3 were possibly homogeneous. Furthermore, the results of HPGPC also validated this. In the results, after eliminating the solvent peak, CCP1 (Figure 6G), CCP2 (Figure 6H), and CCP3 (Figure 6I) all presented symmetrical single peaks.

### 3.4. Analysis of the Results of Determination of Chemical Compositions

The traits and yields of the lyophilized polysaccharides (CCP1, CCP2, CCP3, CCP4, and CCP5) are presented in Figure 7A. As shown, CCP1 and CCP3 were white flocs, while CCP2, CCP4, and CCP5 were white acicular crystals. The yields of the purified polysaccharides were ranked in the following order: CCP1 > CCP2 > CCP5 > CCP3 > CCP4. The overall recovery rate exceeded 80%, suggesting that there was no excessive additional loss during the separation and purification process.

The amounts of total polysaccharides, proteins, and uronic acid of the CCPs were calculated based on the standard curves of glucose (Figure 7B), bovine serum albumin (Figure 7C), and uronic acid (Figure 7D), as presented in Table 6. The data indicated that the carbohydrate content in all five purified polysaccharides exceeded 80%, among which CCP2 showed the highest carbohydrate content at 88.94%. Moreover, all five purified CCPs contained galacturonic acid. CCP3 had the highest uronic acid content at 13.68%, while CCP4 had the lowest at 1.51%, suggesting that all five purified CCPs were acidic polysaccharides. Additionally, trace amounts of proteins were detectable in CCP1 (1.24%) and CCP4 (0.45%), while no proteins were detected in CCP2, CCP3, and CCP5.

The presence of starch in polysaccharides can be identified using the iodine-potassium iodide test. If the solution turns blue, it contains starch; otherwise, it does not [21]. As shown, the color of the solution did not change significantly when the CCPs were mixed with iodine-potassium iodide solution (Figure 7E), suggesting that there was no linear starch-like substance in the CCPs, similar to the result of non-starch polysaccharides from *Colocasia esculenta* peel [27]. Figure 7F demonstrated that the maximum absorption peaks of the purified CCPs were centered around 351 nm, with no absorption peak at around 565 nm, indicating that none of the CCPs contained linear starch but they did contain starches with more branched chains [28]. It could be inferred that in addition to alpha-1,4 glycosidic bonds, the polysaccharides might also have contained alpha-1,6 glycosidic bonds, which have strong stability and cause low irritation to the skin.

### 3.5. Characterization of Polysaccharide Structure

The infrared spectra presented in Figure 8 revealed that each group of purified *Cinnamomum cassia* polysaccharides displayed broad and intense absorption peaks within the 3700–3100 cm^−1^ range, suggesting the existence of O-H stretching vibrations. CCP1 (Figure 8A), CCP2 (Figure 8B), and CCP3 (Figure 8C) all possessed absorption peaks within the 2925–2940 cm^−1^ range, which was associated with the C-H stretching vibrations of methylene or methyl groups in the polysaccharides [29]. Furthermore, within the 1629–1649 cm^−1^ interval, all of the polysaccharides exhibited hydration vibration peaks with similar shapes [30]. Between 1400 and 1600 cm^−1^, the symmetrical and asymmetrical stretching vibrations of carboxyl or carboxylic acid groups lead to the emergence of absorption peaks, indicating that CCP1, CCP2, and CCP3 were all acidic polysaccharides, which was in accordance with them containing uronic acid components [31].

Furthermore, a prominent infrared characteristic absorption peak was witnessed within the range of 1154–1027 cm^−1^, which could be associated with the existence of C-O stretching vibrations in the samples. It is worth noting that characteristic peaks emerged around 890–860 cm^−1^ for all three polysaccharides, suggesting that the samples contained β -glucopyranose rings. Simultaneously, intense characteristic absorption peaks were present in the regions of 783–705 cm^−1^ and 600 cm^−1^, which might be related to the symmetrical ring stretching vibration of glucopyranose [32]. Moreover, absorption peaks were detected in the range of 780–760 cm^−1^ for the samples, indicating that CCP1, CCP2, and CCP3 all incorporated β-mannopyranose rings [33].

These analyses facilitated our comprehension of the fundamental chemical structures of these polysaccharide samples. The basic chemical structure of the cinnamon polysaccharides obtained through microwave extraction was consistent [10]; however, the specific functional groups and structures required further validation by integrating other analytical approaches (such as nuclear magnetic resonance, mass spectrometry, etc.).

### 3.6. Effects of CCPs on Erythrocyte Hemolysis

The safety of the CCPs was initially evaluated using an in vitro hemolysis test. The samples tested were required to meet the standard of hemolysis rate set by ISO 10993-4:2017 [34], which was less than 5% [35]. As shown in Figure 9, CCP5 exhibited the highest hemolysis rate of 4.7%, which was lower than the requirements of ISO. The results showed that the CCPs were safe at a concentration of 3 mg/mL.

### 3.7. Effects of CCPs on Cell Viability of UVB-Induced HaCaT Cells

Figure 10A demonstrates that when the concentration of CCCP, CCP2, CCP3, CCP4, and CCP5 was not higher than 500 μg/mL, the survival rate of HaCaT cells was above 80%. As for CCP1, it did not show cytotoxicity against HaCaT cells when the concentration was below 100 μg/mL. Consequently, a maximum safe concentration of 500 μg/mL was selected for CCCP, CCP2, CCP3, CCP4, and CCP5, while 100 μg/mL was established as the maximum safe concentration of CCP1 in subsequent experimental studies.

As shown, a 15 mJ/cm^2^ UVB irradiation dose did not significantly affect cell viability, while 30 to 120 mJ/cm^2^ UVB irradiation doses significantly impacted cell viability (Figure 10B). The cell viability after a 30 mJ/cm^2^ UVB irradiation dose was about 55%. Ultimately, a 30 mJ/cm^2^ UVB irradiation dose was selected as optimal for modeling.

Photoaging is defined as the aging of human skin resulting from environmental exposure to ultraviolet radiation. When the skin encounters ultraviolet light, it can immediately cause harm to skin cells, altering their structure and function, which ultimately leads to photoaging of the skin. In the present study, human epidermal HaCaT cells were subjected to pretreatment with varying concentrations of the CCPs. The viability of HaCaT cells following UVB exposure was then detected to explore the anti-skin photoaging effect of the CCPs. As shown in Figure 10C, following UVB irradiation, the morphology of cells became abnormal, while the cells treated with the CCPs or hyaluronic acid (HA) remained structurally intact and showed clear and bright cell edges. The survival rate of HaCaT cells in the UVB group was decreased compared with that in the untreated CON group, suggesting that the cells suffered from photoaging damage (Figure 10D). After treatment with 500 μg/mL of HA in the positive group, the survival rate was increased significantly. Nevertheless, after treatment with the CCPs, the HaCaT survival rates were not significantly elevated. These data indicated that HA possessed remarkable anti-photoaging vitality, while the CCPs did not possess anti-photoaging ability.

### 3.8. Effects of CCPs on NO Production in LPS-Induced RAW264.7 Cells

Figure 11A indicated that CCP1 (<100 μg/mL) and CCCP, CCP2, CCP3, CCP4, and CCP5 (<500 μg/mL) did not exhibit significant cytotoxic effects on RAW264.7 cells. Furthermore, it was found that RAW264.7 cells treated with CCCP, CCP1, CCP2, CCP3, CCP4, and CCP5 at the test concentrations maintained good survival rates and cell morphology, remaining transparent with clear edges (Figure 11B). Therefore, the concentration of 500 μg/mL was determined to be the highest safe level for CCCP, CCP2, CCP3, CCP4, and CCP5 and 100 μg/mL was chosen as the highest safe concentration for CCP1 for further experimental studies.

The effects of the CCPs on NO release by RAW264.7 cells are shown in Figure 10C, where DXM was used as a positive control. As shown, NO secretion by RAW264.7 cells in the LPS group was notably greater compared to that in the CON group, suggesting that the inflammation of RAW264.7 cells was stimulated by LPS. However, following treatment with DXM and the CCPs, NO production by RAW264.7 cells significantly decreased in a dose-dependent manner. Notably, the NO content in the polysaccharide-treated groups was close to that in the untreated normal cells (Figure 11C). The inhibitory effects of the CCPs on NO production were in the following order: CCCP > CCP1 > CCP4 > CCP5 > CCP3 > CCP2 > DXM. These facts suggested that the CCPs had anti-inflammatory potential.

## 4. Conclusions

The GA-BP neural network model was used to predict the optimal extraction condition of CCCP, which was as follows: cellulase was employed, the temperature for enzymatic hydrolysis was 50.0 °C, the pH during the process was 5.248, the addition of enzyme was 3%, and the ultrasonic time was 70.153 min. The theoretical yield was 28.42% and the actual yield was 28.35%. Further purification of CCCP using DEAE-52 and Sephadex G-100 columns indicated that fractions CCP1, CCP2, and CCP3 were possibly homogeneous. Hemolysis experiments demonstrated that the hemolysis rate of the CCPs at 3 mg/mL was lower than 5%, indicating their safety. It was revealed that the CCPs did not exert anti-photodamage effects on UVB-irradiated HaCaT cells. However, the CCPs markedly reduced NO release in RAW264.7 cells stimulated by LPS, with CCCP and CCP1 being more effective. In conclusion, this study offers a theoretical foundation for the future use of the CCPs in treating inflammation-related diseases.

## Figures and Tables

**Figure 1 foods-14-00686-f001:**
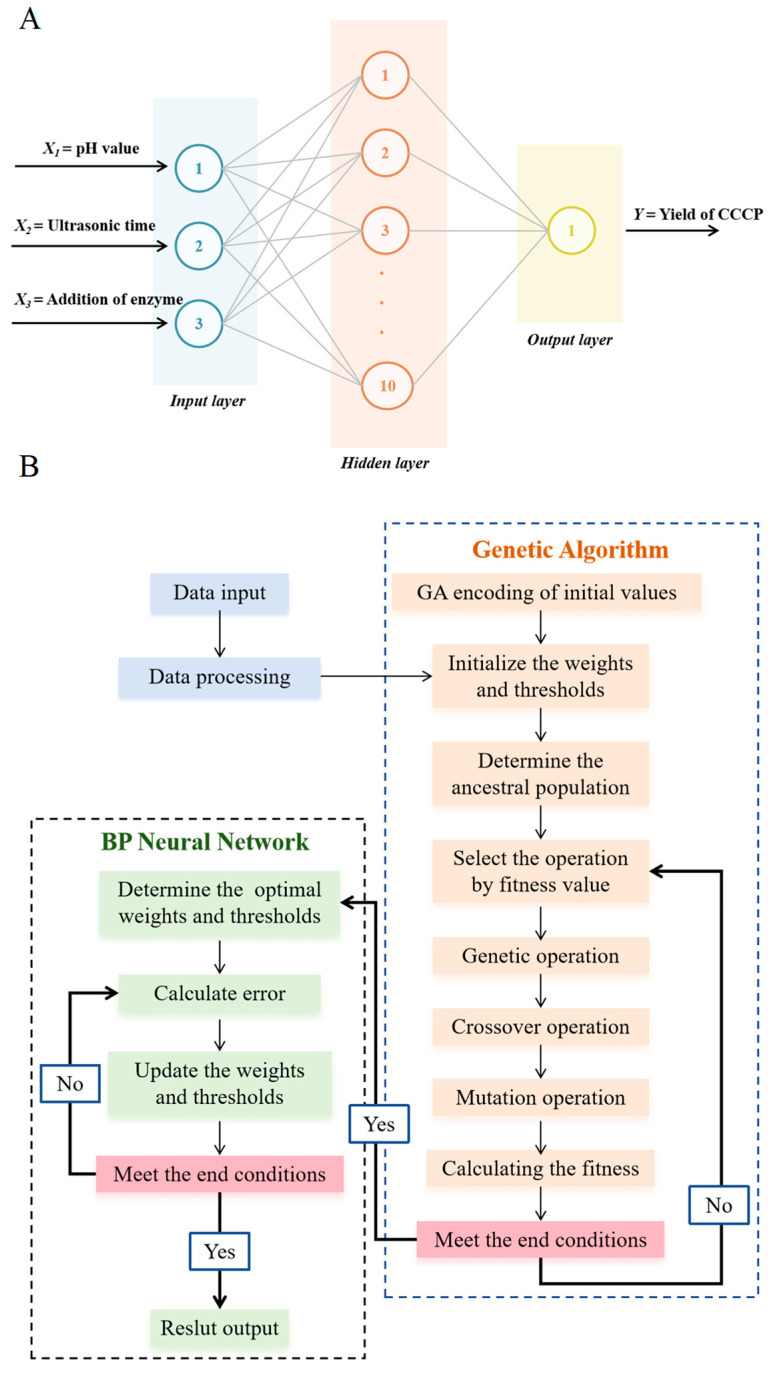
The optimal architecture of the ANN model. (**A**) BP-NN model architecture topology (3–10–1). (**B**) Flowchart of GA-BP algorithm.

**Figure 2 foods-14-00686-f002:**
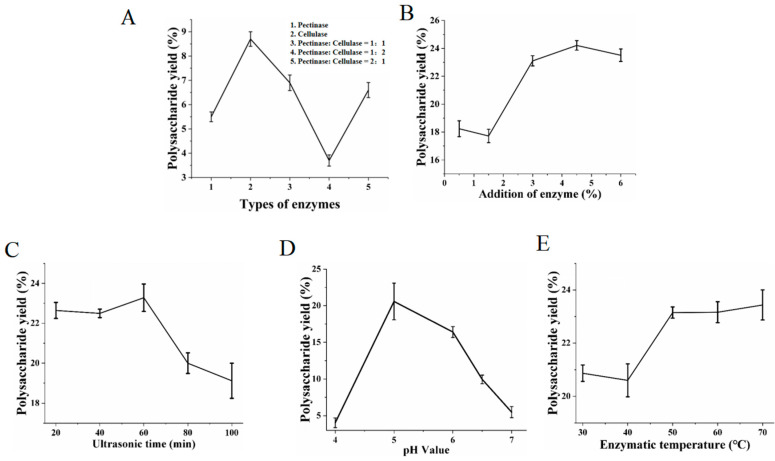
The yield and recovery of CCCP extracted from different single-factor experiments: (**A**) types of enzyme; (**B**) addition of enzyme; (**C**) ultrasonic time; (**D**) pH value; (**E**) enzymatic temperature.

**Figure 3 foods-14-00686-f003:**
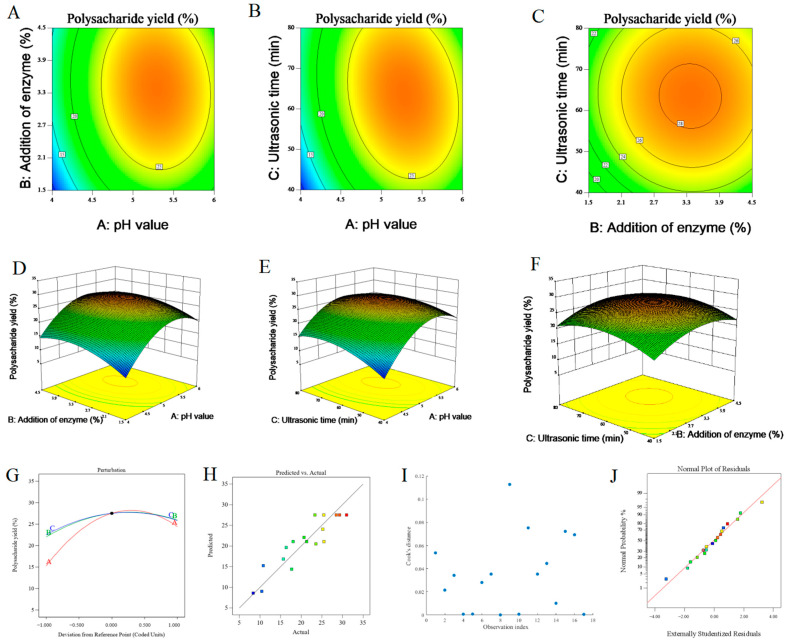
RSM for optimizing the UAEE of CCCP. (**A**–**C**) Contour maps illustrating the interactive effects of pH value, addition of enzyme and ultrasonic time on the yield of polysaccharides; The colors in this plot likely represent different levels of yield variable, with cooler colors (blues) indicating lower values and warmer colors (reds, yellows) indicating higher values. (**D**–**F**) the corresponding three-dimensional response surface plots; (**G**) perturbation graph presenting the experimental design space, where A stands for pH value, B for addition of enzyme, and C for ultrasonic time; (**H**) comparative graph of actual values and model predicted values; (**I**) Cook’s distance; (**J**) normal probability.

**Figure 4 foods-14-00686-f004:**
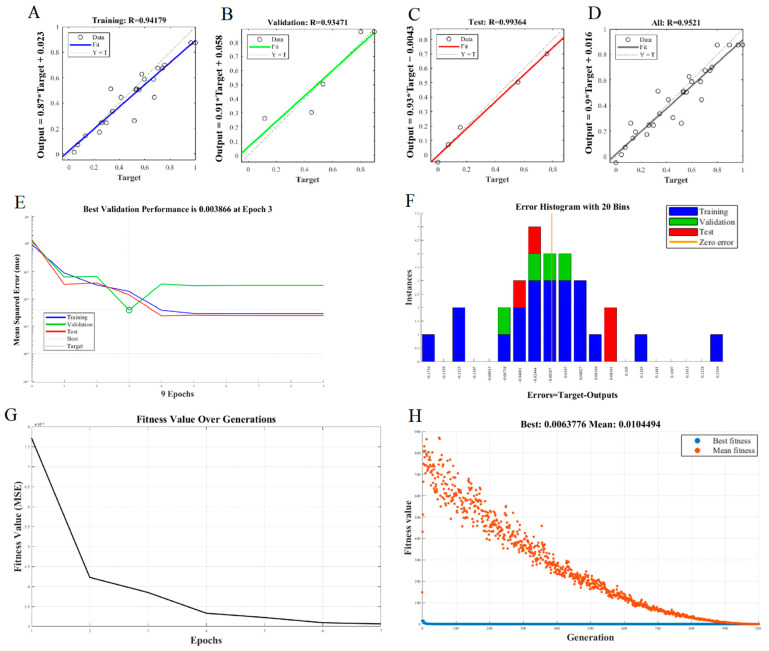
The model performance verification and analysis: scatter plot between experimental and predictive data by ANN modeling for (**A**) training, (**B**) testing, (**C**) validation, and (**D**) overall data fitting; (**E**) performance; (**F**) error histogram. (**G**) The variation diagram of fitness values based on multiple generations and (**H**) the comparison diagram of the best and average fitness values.

**Figure 5 foods-14-00686-f005:**
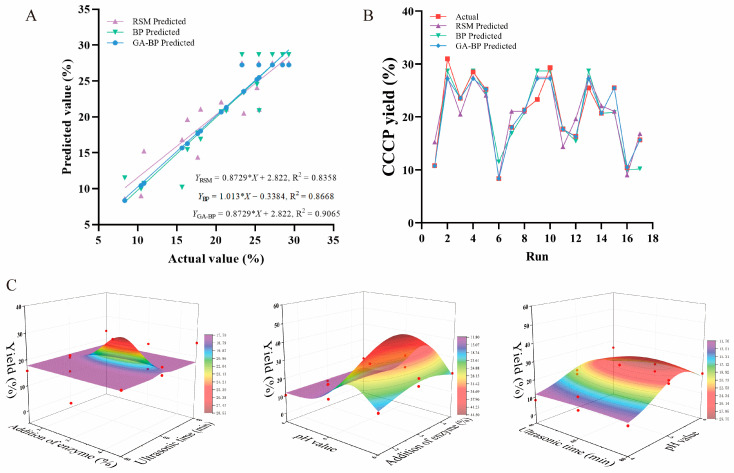
Comparison of prediction capabilities of RSM, BP and GA-BP neural network models. (**A**) Fitting of model predicted values with experimental values, (**B**) matching between all of the datasets, and (**C**) neural network 3D surface plot of the three independent variables to the interaction of CCCP yield.

**Figure 6 foods-14-00686-f006:**
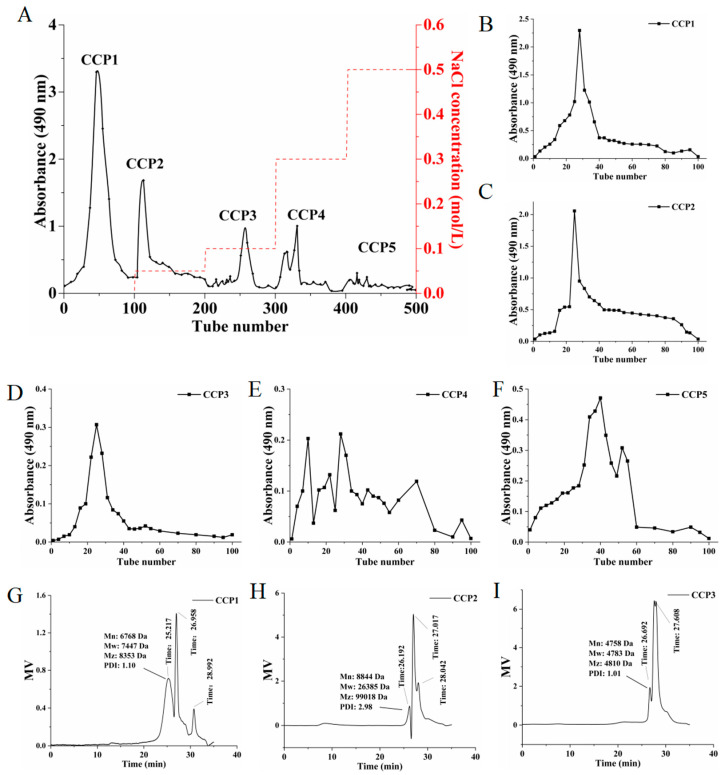
Isolation and identification of purified polysaccharides from cinnamon. (**A**) DEAE-52 column chromatography elution profiles of CCCP and Sephadex G-100 column chromatography elution profiles of (**B**) CCP1, (**C**) CCP2, (**D**) CCP3, (**E**) CCP4, and (**F**) CCP5. The HPGPC results of (**G**) CCP1, (**H**) CCP2, and (**I**) CCP3.

**Figure 7 foods-14-00686-f007:**
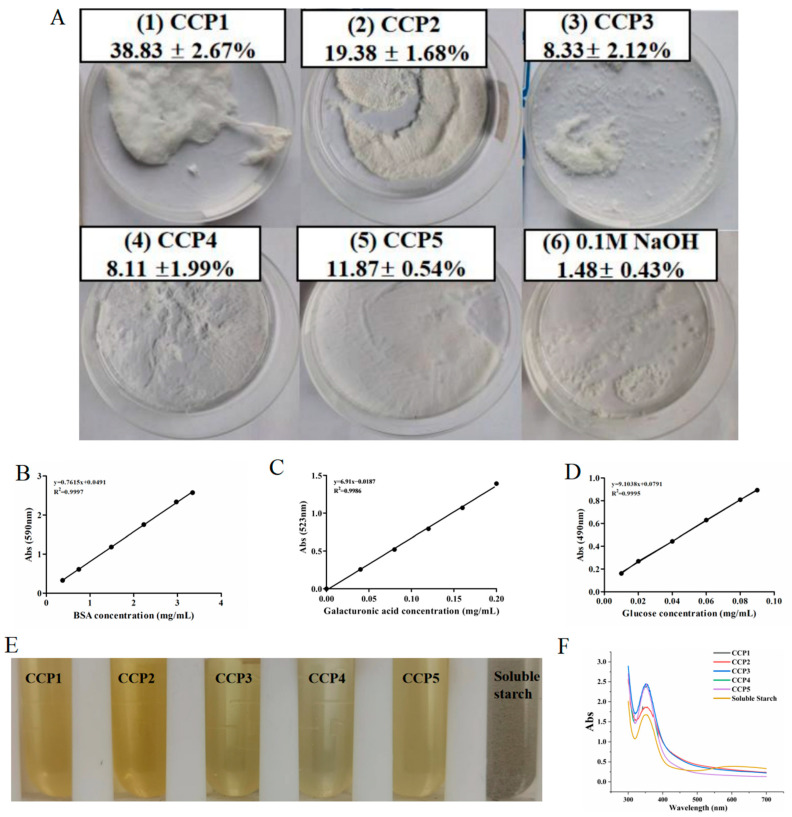
Physicochemical properties and chemical compositions of CCPs. (**A**) Characteristics and yield of CCPs. Standard curves of (**B**) total carbohydrate, (**C**) protein, and (**D**) uronic acid. (**E**) Visual representation and (**F**) ultraviolet scanning results of I_2_-KI test on CCPs.

**Figure 8 foods-14-00686-f008:**
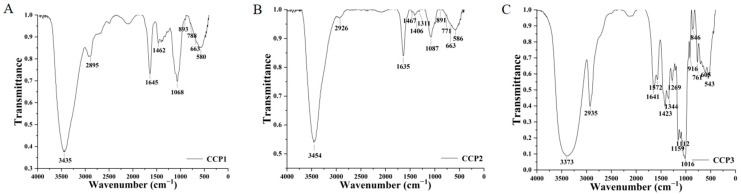
Infrared spectroscopy analysis diagrams of (**A**) CCP1, (**B**) CCP2, and (**C**) CCP3.

**Figure 9 foods-14-00686-f009:**
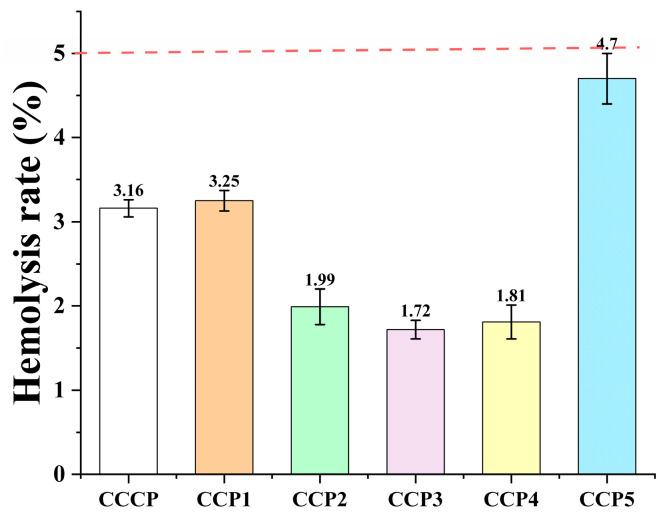
Hemolysis rate of CCPs.

**Figure 10 foods-14-00686-f010:**
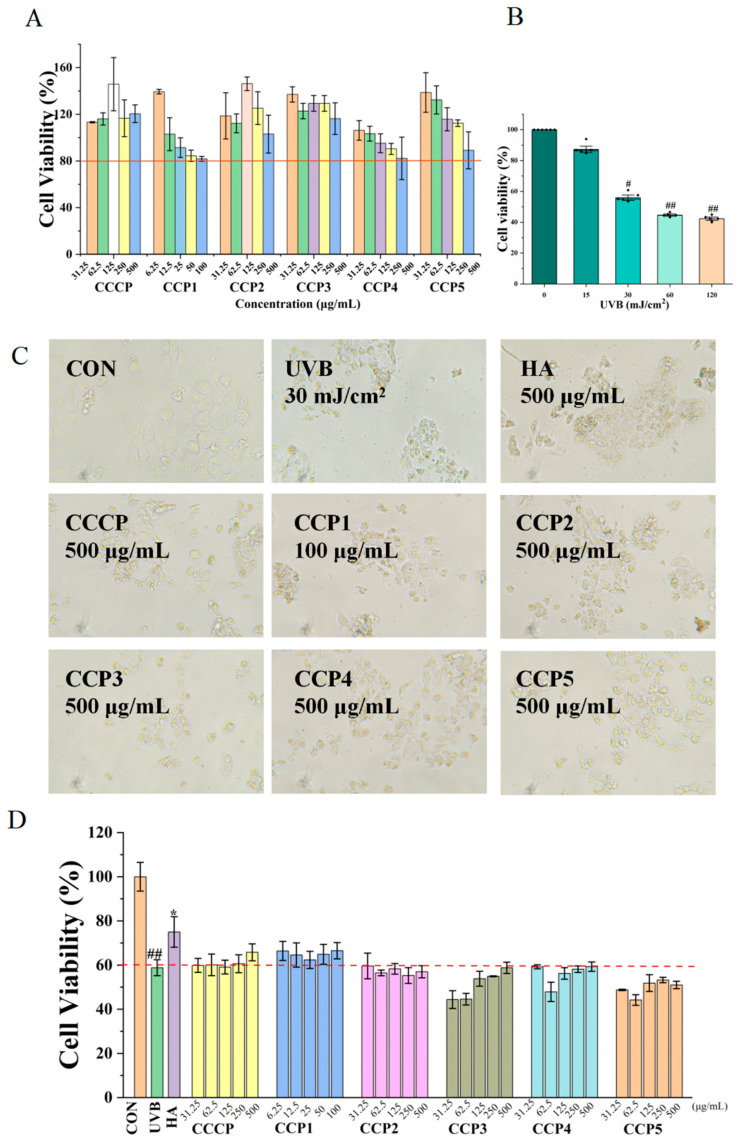
Effects of CCPs on cell morphology and viability of UVB-induced HaCaT cells. Effects of (**A**) CCPs and (**B**) UVB radiation on cell viability of HaCaT cells. Effects of CCPs on (**C**) cell morphology and (**D**) cell viability of UVB-induced HaCaT cells. * *p* < 0.05, vs. UVB group, ^#^
*p* < 0.05, ^##^
*p* < 0.01, vs. CON/0 group.

**Figure 11 foods-14-00686-f011:**
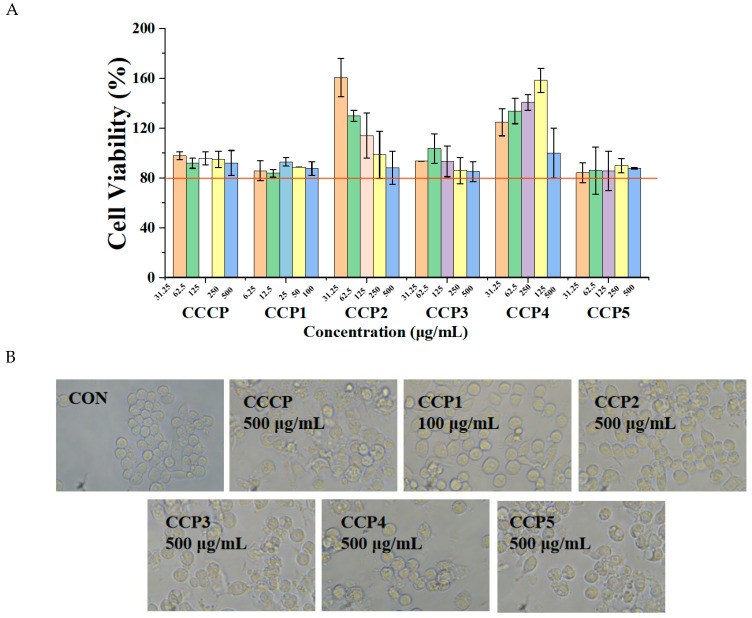
Effects of CCPs on cell morphology and NO content in LPS-induced RAW264.7 cells. Effects of CCPs on (**A**) cell viability, (**B**) cell morphology (200×), and (**C**) NO production in LPS-induced RAW264.7 cells. ^###^
*p* < 0.001, vs. CON group. * *p* < 0.05, ** *p* < 0.01, *** *p* < 0.001, vs. LPS group.

**Table 1 foods-14-00686-t001:** The factors and levels of RSM.

	Levels		
	−1	0	1
A: pH value	4	5	6
B: Addition of enzyme (%)	2	3	4
C: Ultrasonic time (min)	20	40	60

**Table 2 foods-14-00686-t002:** The actual and predicted data from the RSM.

Std	A: pH Value	B: Addition of Enzyme (%)	C: Ultrasonic Time (min)	Yield (%)			
				Actual	RSM Predicted	BP Predicted	GA-BP Predicted
1	4	1.5	60	10.44	9.035	9.98	10.45
2	6	1.5	60	16.36	19.645	15.45	16.28
3	4	4.5	60	17.68	14.395	17.76	17.67
4	6	4.5	60	20.68	22.085	20.72	20.68
5	4	3	40	8.36	8.6	11.52	8.36
6	6	3	40	25.52	21.07	20.86	25.49
7	4	3	80	10.8	15.25	10.73	10.75
8	6	3	80	21.32	21.08	20.83	21.32
9	5	1.5	40	15.68	16.845	10.20	15.68
10	5	4.5	40	18.04	21.085	16.90	18.04
11	5	1.5	80	23.56	20.515	23.37	23.55
12	5	4.5	80	25.24	24.075	24.53	25.24
13	5	3	60	29.31	27.522	28.68	27.24
14	5	3	60	28.5	27.522	28.68	27.24
15	5	3	60	25.48	27.522	28.68	27.24
16	5	3	60	31	27.522	28.68	27.24
17	5	3	60	23.32	27.522	28.68	27.24

**Table 3 foods-14-00686-t003:** RSM design and corresponding response values.

Source	Sum of Squares	df	Mean Square	F Value	*p*-Value
Model	614.36	9	68.26	3.84	0.0451 *
A	167.45	1	167.45	9.41	0.0181 *
B	30.42	1	30.42	1.71	0.2324
C	22.18	1	22.18	1.25	0.3011
AB	2.13	1	2.13	0.1198	0.7395
AC	11.02	1	11.02	0.6193	0.4571
BC	0.1156	1	0.1156	0.0065	0.938
A^2^	248.41	1	248.41	13.96	0.0073 ***
B^2^	53.09	1	53.09	2.98	0.1278
C^2^	47	1	47	2.64	0.1482
Residual	124.59	7	17.8		
Lack of Fit	86.51	3	28.84	3.03	0.1561
Pure Error	38.08	4	9.52		
Cor Total	738.95	16			

* *p* < 0.05, *** *p* < 0.001.

**Table 4 foods-14-00686-t004:** Comparison of prediction capabilities of RSM, BP, and GA-BP neural network models.

Model	RSM	BP	GA-BP
R^2^	0.8358	0.8668	0.9065
RMSE	2.8678	2.0423	1.6215
AAD	2.26	1.99	1.12

**Table 5 foods-14-00686-t005:** The yield of extraction predicted by RSM, BP, and GA-BP neural network models with optimal conditions and compared with experimental data. The value in parentheses in the “pH value” column is the actual experimental pH value.

Model	Optimal Conditions			Yield (%)		
	A: pH Value	B: Addition of Enzyme (%)	C: Ultrasonic Time (min)	Predicted	Actual	MSE
RSM	5.997 (6.00)	3.825	79.895	27.95	28.54	0.3410
BP	5.287 (5.29)	3.801	59.269	27.71	28.12	0.1616
GA-BP	5.248 (5.25)	3	70.153	28.42	28.35	0.0049

**Table 6 foods-14-00686-t006:** The chemical composition of CCPs.

Samples	Total Carbohydrate (%)	Protein (%)	Uronic Acid (%)
CCCP	72.41	0.52	7.91
CCP1	80.63	1.24	11.57
CCP2	88.94	/	5.10
CCP3	80.74	/	13.68
CCP4	84.58	0.45	1.51
CCP5	86.53	/	9.85

## Data Availability

The original contributions presented in this study are included in the article. Further inquiries can be directed to the corresponding author.

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
