# Peer review of "Genetic Algorithm-Back Propagation Neural Network Model- and Response Surface Methodology-Based Optimization of Polysaccharide Extraction from Cinnamomum cassia Presl, Isolation, Purification and Bioactivities"

_foods, 2025, doi:10.3390/foods14040686_

Round 1

Reviewer 1 Report

Comments and Suggestions for Authors

The study presented in the manuscript titled “Genetic algorithm-back propagation neural network model and response surface methodology-based optimization of polysaccharide extraction from Cinnamomum cassia Presl, isolation, purification and bioactivities” is relevant as it utilizes the GA-BP model to predict the optimal conditions for polysaccharide extraction from Cinnamomum bark. However, the authors are requested to address the following major concerns:

1.       Line 72-74: Normally, ultrasonic-assisted extraction and subcritical extraction have never been classified as conventional extraction methods, but rather as unconventional or modern methods, alongside microwave extraction, etc. Please correct this.

2.       Line 92-92, please specify if the ethanol precipitation of the polysaccharides was carried out at room temperature or not.

3.       Line 101-103: The authors state “Each experiment concentrated on changing a single factor while maintaining the other variables at a constant level.” The authors should specify the constant values of each variable used in each experiment.

4.       Line 157: The term "total polysaccharide content" should be replaced with "total carbohydrate content" throughout the manuscript, as the analysis with the phenol-sulfuric method quantifies total carbohydrates or sugars.

5.       Line 170, please define what MTT is. Were the fractions and CCCP dissolved in DMSO or water to carry out the cell viability and NO assays? If so, have the authors considered the effects of this on the biological activities of the samples?

6.       Line 393: According to the yields in Figure 7A, the overall recovery rate is around 85%. How can the loss of approximately 15% of the mass be explained? Furthermore, was the yield calculated based on the mass of the crude CCCP extract or that obtained after ethanol precipitation?

7.       Line 387-390, upon examining Figures 6B, C, D, etc., it is difficult to state that fractions CCP1, CCP2, and CCP3 exhibit a single and symmetrical peak and are therefore homogeneous. The authors should support/confirm these results, for instance, with HPGPC analysis.

8.       Line 422, the chemical compositions of the CCCP (crude extract) should also be presented.

9.       Line 481/193: Have you evaluated the possible LPS contamination in CCCP and its fractions? This could affect the results.

10.   There is confusion in the use of the abbreviations "CCPS" and "CCPP." Does CCPs also include CCPP? This needs to be clarified.

Comments on the Quality of English Language

The manuscript is well written

Reviewer 2 Report

Comments and Suggestions for Authors

Dear authors,

Here are my  questions and comments regarding your manuscript:

1. Lines 38-40 "As macromolecular components, polysaccharides are typically extracted by the water extraction method due to their property of soluble in water but insoluble in ethanol" . There are also other methods to extract polysaccharides  from biomass. You should mention them.

2. What was the chemical composition of raw material Cinnamomum cassia Presl (C. cassia)? in terms of major and minor components ... cellulose hemicelluloses  lignin.. extractives and protein ? The method used for carbohydrates determination is not specific, so the samples must be re-analyzed (https://link.springer.com/chapter/10.1007/978-1-4419-1463-7_6) and obviously the discussion and conclusions section has to be modified.

The detailed conditions for protein content in raw material determination should be mentioned in materials and methods.

3. My guess is that the extracted polysaccharides belong to the hemicelluloses group but there is no mention about this. So I recommend a discussion on the structure of the extracted polysaccharides (from) Cinnamomum cassia Presl (C. cassia) this  should be based on literature existing studies. On the other hand a chemical structure study (based on experimental data) should be included to reveal the the characteristics of your samples.

4. Please specify the source and type of  D354FD resin.

5. The sonication equipment should be presented. The operating frequency needs to be specified.

6. The relationship for the calculation of yield of extraction needs to be mentioned.

7.  The supplier for DEAE-52 column (3.5 × 50 cm) is not mentioned.

8. Table 5 specifies several optimal parameters. The theoretical optimal pH was mentioned with 3 digits in each case. In reality it is very difficult to measure and to maintain the pH in such manner and precision . Most of the instruments only display 2 decimas digits. Therefore, the real experimental pH values should be also included.

Reviewer 3 Report

Comments and Suggestions for Authors

Minor remarks

-        Latin terms should be presented in italics. Please, check the whole manuscript including the references list.

-        Please, provide a blank space between quantity and unit except in the case of percentage.

-        Line 46: After “Jong Jin Park et al.” insert the reference “[8]” and delete the number at the end of the sentence. Please, correct it throughout the text.

-        Instead of minutes and hours, use the unit “min” and “h”.

-        Ortho, meta, para- in the abbreviated form should be presented in italics.

-        Line 202: Instead of “#p values”, insert “p-values”.

-        Line 264: delete one “in”.

Major remarks

-        Avoid lumping the references without separately discussing them in detail. For instance, Lines 58-89: “The optimization of polysaccharide extraction processes mainly adopts the response 58 surface method (RSM), etc [1, 5, 9-10].

-        In Table 1, the factor levels of pH value are not adequate and should be retyped. The units of used factor levels should also be inserted.

-        Generally, the table and figure captions should be more informative.

-        Line 118: You mention in the manuscript the following statement “one hidden layer (comprising ten hidden layer neurons),”. You should know that in many cases, a larger number of neurons can lead to overfitting the model to a small dataset, which can reduce its ability to generalize to new data. Essentially, the number of neurons in the hidden layer should be adapted to the complexity of the problem as well as the size of the dataset. For small datasets, simpler models with fewer neurons usually yield better results.

-        Line 120-121: The following sentence “The complete dataset from 17 extraction experiments was divided into three subsets.” is questionable. This sample size is too small for model generation using an artificial neural network. Additionally, it should be noted that the center point of the Box-Behnken design was repeated multiple times. These repeated values should be averaged and presented as a single value. More extractions are needed for a robust model. Given that 15% of the data corresponds to approximately 2.5 extractions, a minimum of 5 extractions per test and validation process is required for the neural network model. Therefore, a total of 23 extractions would be necessary for testing. I would like to emphasize again that the extractions of the center point should be averaged. The total number of extractions required is 23 + 5 + 5 = 33 extractions.

-        In section 2.6, provide more detail about the used equipment (model, manufacturer, city, state, country).

-        In Table 2, the extraction of the design center point should be indicated to be visible. Also, the used data for training, testing, and validation of neural networks need to be indicated.

-        Please, indicate all unknown symbols in the footnotes of the table.

-        In the caption of Figure 3, more detail of the plots of A, B, C, D, E, F, G, and H should be described in detail. The caption should be expanded to include these thorough explanations. Also, plots of normal probability and Cook's distance should be provided because they are more significant than the plot predicted vs actual values.

-        After analysis of Figure 4A-D, it looks like you used 5 experiments to test and validate neural networks. How is it possible, if you have a total data of 17? Can you explain that?

Comments on the Quality of English Language

English is acceptable, but some grammatical errors need to be improved.

Round 2

Reviewer 1 Report

Comments and Suggestions for Authors

Authors have adequately addressed all of my concerns. Therefore, I recommend the publication of the manuscript in its current version.

Author Response

We sincerely thank you for your meticulous review of our manuscript and your valuable suggestions.  Your professional opinions are of great significance to us in improving our research work.

Reviewer 2 Report

Comments and Suggestions for Authors

Dear authors, 

I agree with most of the explanations and modifications. However, the discussion regarding the chemical structure of the extracted should be extended based on existing literature data (some examples: https://www.sciencedirect.com/science/article/pii/S0141813022029178, Silva, Fernanda & Nunes, Cláudia & Coimbra, Manuel. (2017). Structural characterization of cinnamon polysaccharides submitted to different hydrolysis treatments. 10.13140/RG.2.2.16078.72005). I believe others may be found too. 

A FTIR study on the probably existing samples will most likely improve your work in absence of other data.

Author Response

Thank you for your suggestions. The manuscript has been revised. Please see the attachment.

Reviewer 3 Report

Comments and Suggestions for Authors

In Figure 6a, the y-axis is not visible.

The sentence in lines 430-431, should be given in the table caption.

Author Response

(The authors gave the same response as above.)
